# Strengthening Mechanism of Electrothermal Actuation in the Epoxy Composite with an Embedded Carbon Nanotube Nanopaper

**DOI:** 10.3390/nano11061529

**Published:** 2021-06-09

**Authors:** Petr Slobodian, Pavel Riha, Robert Olejnik, Jiri Matyas

**Affiliations:** 1Centre of Polymer Systems, University Institute, Tomas Bata University, Tr. T. Bati 5678, 760 01 Zlin, Czech Republic; olejnik@utb.cz (R.O.); matyas@utb.cz (J.M.); 2Faculty of Technology, Polymer Centre, Tomas Bata University, T.G.M. 275, 760 01 Zlin, Czech Republic; 3The Czech Academy of Sciences, Institute of Hydrodynamics, Pod Patankou 5, 166 12 Prague 6, Czech Republic

**Keywords:** carbon nanotube nanopaper, epoxy composite, Joule heating, glass transition, actuation force

## Abstract

We assessed an effect of an embedded electro-conductive multiwalled carbon nanotube nanopaper in an epoxy matrix on the release of the frozen actuation force and the actuation torque in the carbon nanotube nanopaper/epoxy composite after heating above its glass transition temperature. The presence of the nanopaper augmented the recovery of the actuation stress by the factor of two in comparison with the pure epoxy strips. We proposed a procedure that allowed us to assess this composite strengthening mechanism. The strengthening of the composite was attributed to the interlocking of the carbon nanotubes with the epoxy. When reheated, the composite samples, which contained stretched mutually intertwined nanotubes and epoxy segments, released a greater actuation stress then the epoxy samples, which comprised of less elastic networks of crosslinked segments of pure epoxy.

## 1. Introduction

A carbon nanotube (CNT) nanopaper, also called buckypaper, is an aggregate of entangled carbon nanotubes (CNTs) produced by a filtering of a CNT suspension onto a membrane support. After drying, the nanopaper is removed from the support, leaving a free-standing electrically and thermally conductive structure. Properties of the nanopaper can be tailored by a selective attachment of organic or inorganic moieties to CNT surfaces [1]. Such modified nanopapers can be used for monitoring of an extensional and compressive deformation [2,3], vapor and gas adsorption [4,5], glucose level [6], liquid penetration [7,8] or be used as photovoltaic electrodes [9], etc. Versatility of nanopapers can be extended even further by embedding it into various materials. Such composites can be used as motion sensors [10,11], electro- and thermo-mechanical actuators [12], antenna-based gas sensors [13], adhesive films [14], thermoelectric devices [15], or built-in sensors for monitoring of material structural health [16,17].

Recently, we have combined a multiwalled carbon nanotube nanopaper and an epoxy resin into a reinforced thermosetting polymer matrix composite and demonstrated its versatile use for the self-regulation of the epoxy curing temperatures as well as for the fast and efficient debonding of objects glued by this MWCNT nanopaper/epoxy composite [18]. Further, our previous results have suggested a considerable shape memory potential of these nanocomposites [18].

The shape memory in cured epoxies and their composites has been ascribed to an interlocking of molecular segments of the epoxy and the embedded filler materials. When such a composite is heated above its glass transition temperature *T*_g_ and subjected to an external load, bonding between its components rearrange to adapt to the external load. Such an altered configuration is locked when the composite cools below its *T*_g_, yet, once the composite is re-heated above its *T*_g_, the original shape is fully or partially recovered [19]. The mechanisms affecting the shape memory in epoxy composites with embedded carbon nanotube nanopaper have not been described as yet. In this paper, we assessed how an electro-conductive multiwalled carbon nanotube nanopaper embedded in an epoxy matrix modulated the release of the frozen actuation force and the actuation torque from the carbon nanotube nanopaper/epoxy composite after its heating above its *T*_g_. We proposed a procedure, which allowed measuring of the strengthening of the composites and analyzing of the electrothermal actuation controlled by Joule heating. By this method, we determined that the presence of the electro-conductive multiwalled carbon nanotube nanopaper increased the actuation stress and the actuation torque two-fold in comparison with the pure epoxy.

## 2. Experimental

### 2.1. Materials

Purified MWCNTs were supplied by Sun Nanotech Co. Ltd., China. According to the supplier, the nanotube diameter was 10–30 nm, the length 1–10 μm, the purity ~90% and the volume resistivity 0.12 Ωcm. The diameter of individual nanotubes was between 10 and 60 nm. The length was from 0.2 μm up to 3 μm. The nanotube consisted of 15 to 35 rolled layers of graphene and the interlayer distance was ca. 0.35 nm [4,20].

The aqueous dispersion of MWCNTs containing 0.8 mg of nanotubes, 530 mL of water with a surfactant system based on a solution of 15.4 g sodium dodecyl sulphate as a surfactant agent and 8.5 mL of 1-pentanol as a co-surfactant agent was prepared by sonication using the Dr Hielscher GmbH UP400St apparatus (ultrasonic horn S7, amplitude 88 μm, power density 300 W/cm^2^, frequency 24 kHz) for 15 min at 50% power of the apparatus, 50% pulse mode at the temperature of about 50 °C. Subsequently, an aqueous solution of NaOH was added to adjust pH to 10.

To make a MWCNT nanopaper from pristine nanotubes, the nanotubes were deposited on a porous polyurethane electrospun non-woven membrane by a vacuum filtration. Two hundred and fifty milliliters of the homogenized MWCNT dispersion was filtered through a funnel of the diameter 90 mm. The resulting disk-shaped filtration cake was washed in situ several times with deionized water (at 65 °C) and afterwards with methanol. In turn, the filtering membrane was peeled off and the filtration cake was dried between two filtration papers for 24 h at room temperature. The measured electrical conductivity of the resulting MWCNT nanopaper was 11.97 S/cm. The thickness of the nanopaper was around 300 μm.

The two-component epoxy resin Epox G 200 (Davex Chemical s.r.o., Prague, Czech Republic) was a transparent epoxy casting system with adjustable hardness and extended processing time. The component A was an epoxy resin prepolymer (hydrogenated bisphenol A polymer with epichlorohydrin [CAS: 30583-72-3]). The prepolymer was prepared with a sufficient excess of epichlorohydrin, so the termination was by free oxirane rings, which were capable of reacting chemically with amine groups. The component B (Trimethylolpropane tris [poly(propylene glycol), amine terminated] ether [CAS: 39423-51-3]) was a hardener or a curing agent containing three polymeric chains of poly(propylene glycol) terminated by amine groups. Structural formulae of components are shown in Figure 1. 

To prepare the electroconductive MWCNT nanopaper/epoxy composite strip (40 mm × 20 mm), at first were two Cu strip electrodes glued at the opposite sides of the about 300 μm-thick nanopaper. Next, the porous nanopaper was laid on a polytetrafluorethylene (PTFE) foil and its pores were filled up with ca. 0.2 mL of the above described epoxy resin and curing agent at the A:B ratio 100:75. After filling, the nanopaper permeated by the epoxy resin and curing agent mixture was covered with another PTFE foil and loaded with a pressure of 300 kPa to avoid wrinkling in the course of curing by Joule heating at temperature around 60 °C for 15 min. The measured electrical conductivity of the resulting MWCNT nanopaper/epoxy composite was 3.16 S/cm. The glass transition temperature of the epoxy matrix with the embedded nanopaper, measured as the peak temperature by the DSC, was 53.9 °C.

### 2.2. Methods of Measurements

The structure of the MWCNT nanopaper and the cross-section of the nanopaper/epoxy composite were analyzed by the scanning electron microscope (SEM) Nova NanoSEM 450 (FEI Co., Lincoln, NE, USA). The surface temperature of the specimens was measured by means of the thermal camera Flir E5-XT (FLIR Systems, Inc., Wilsonville, OR, USA), which was able to create a detailed temperature pattern.

The sensor resistance was measured lengthwise by the two-point technique using the fast-scanning data logger Multiplex datalogger 34980A (Keysight Technologies, Santa Rosa, CA, USA), which stored the readouts once per second. The DC power supply AX 502 (AEMC Instruments, Dover, NH, USA) was used to power Joule heating of the composites. Tensile tests were carried out using the Testometric M350-5CT system (The Testometric Co. Ltd., Rochdale, UK).

Two different tests of a strengthening mechanism of electrothermal actuation were done to assess the role of an embedded MWCNT nanopaper. In the first test, the stretched MWCNT nanopaper/epoxy composite strips and the epoxy strips were heated above their respective *T*_g_, and then let to cool down at ambient temperature while subjected to a given tensile strain. Afterwards, the stretched glassy strips were reheated to the rubbery state while measuring an increase of the actuation force in the strips. The composite strips were heated by Joule heating and the epoxy strips by warm air. In the former, two electrodes were attached lengthwise to the opposite sides of the strip. These electrodes were made of Cuprexit (a thin copper layer supported by a 0.22 mm thick fiberglass plate) to which conductive wires were soldered. In the latter, a hot air blower Triac PID (Leister Technologies AG, Sarnen, Switzerland) was used for heating. The distance of the blower from the epoxy strip was about 30 cm and the temperature of the outgoing air was 200 °C. In both cases, the temperature was measured in the center of the strip by a thermal camera as well as times necessary for reaching of given temperature values.

The second test assessed a straightening of bent strips, which was induced by Joule heating in the MWCNT nanopaper/epoxy composite samples or by warm air in the pure epoxy. The composite and epoxy strip were warmed about 12 °C above their *T*_g_, i.e., to 65 °C and 47 °C, respectively. At these temperatures, the samples went from being rigid and glassy to being rubbery, flexible, and highly formable. When a bending force was applied, the strips were easily bent by hand at right angles around a cylinder with a 5 mm diameter. The altered shape held after the strips had cooled down to room temperature. The bent strips were attached to a pad by a double-sided tape, heated again to 65 °C and 47 °C, respectively, and fully or partially straightened. The straightening angle was measured visually by a comparison with a scale.

## 3. Results

The poly(propylene glycol) chains themselves are very movable, capable to rotate around a single chemical bond. The chains contain oxygen, which has a relatively low rotational barrier and therefore the conformation of the chains can change easily under, e.g., an applied mechanical stress. The reaction of the two components A and B leads to a formation of a three-dimensional polymer network of a final epoxy matrix. At the A:B ratio 100:100, an excess of the component B causes that some poly(propylene glycol) branches remain unbounded because of a lack of oxirane rings with a high molecular mobility potential, and the epoxy is elastic. At the ratio 100:50, the proportion of functional groups for crosslinking between components A and B is more equimolecular. The epoxy contains less free unbounded poly(propylene glycol) branches, which results in a resistance against conformation changes, stiffness and a higher *T_g_*. According to the supplier, the ratio of the epoxy components (A/B-hard/elastic) 100:50 yields a hard epoxy of hardness 79 Shore D, while the component ratio of 100:100 yields a flexible one (44 Shore A). The glass transition temperatures *T*_g_ of the epoxy with component ratio 100:100, 100:75 and 100:50 were determined by the differential scanning calorimetry (DSC) analysis (DSC 1, Perkin Elmer, Waltham, MA, USA) to be 18.4, 35.1 and 58.1 °C, respectively. The chosen mixing epoxy component ratio of 100:75 enabled that once the experimental epoxy samples were heated above their *T*_g_ of 35.1 °C, they were able to be formed manually. The curing time of such experiments samples was 48 h at room temperature. The exothermic heat of the curing was determined by the DSC analysis as −275 J/g and the peak temperature 140 °C at the heating rate 10 °C/min. The cured epoxy was elastic and flexible (without any damage at bending over radius 70 mm), its ultimate tensile strain was 3.45% and hardness 66 Shore D.

A tensile test was carried out to determine the fracture tensile stress and the strain of the MWCNT nanopaper/epoxy composite and the cured epoxy at the temperature 75 °C, which was above their respective *T*_g_ temperatures. The composite elongated with strain up to the fracture stress of 2.4 MPa at the strain 5.5% and the cured epoxy up to the fracture stress of 0.95 MPa at the strain 7.5%. The strain dependency of stresses for the rubbery MWCNT nanopaper/epoxy composite and the epoxy was linear and the Young’s moduli, which are defined as *E* = σ/*ε* (where σ denotes tensile stress and *ε* axial strain) were 46 MPa and 24 MPa, respectively.

Similar data taken for illustration from our paper [18] for the temperature of 25 °C, which was below their respective *T*_g_ temperatures, are shown also in Figure 2. The MWCNT nanopaper broke sharply at the fracture stress 1 MPa and the strain 0.75%. The glassy cured epoxy experienced at first a strain hardening through a plastic deformation and then a necking owing to the plastic deformation until the fracture stress of 13.5 MPa and the strain 3.45%. On the other hand, the MWCNT nanopaper/epoxy composite cured by Joule heating at 4.4 V and the terminal temperature of 61 °C strengthened until the fracture stress of 24 MPa at the strain 2.9% as seen in Figure 2. The epoxy matrix increased the fracture strain of the glassy MWCNT nanopaper/epoxy composite almost four times compared to the MWCNT nanopaper. The Young’s modulus of the cured epoxy, the MWCNT nanopaper/epoxy composite and the MWCNT nanopaper was 1.15 GPa, 1.15 GPa and 133 MPa, respectively.

Next, we assessed a strengthening mechanism of the electrothermal actuation in the MWCNT nanopaper/epoxy composite. The extended MWCNT nanopaper/epoxy composite and the epoxy strips were cooled down from 75 °C to the ambient temperature of 25 °C and then reheated to the rubbery state at 75 °C at the rate 2.7 °C/s and 3.3 °C/s for the composite and the pure epoxy strips, respectively. When the strips were heated again, the stress response in the pre-deformed strips initially began to decrease due to the thermal expansion of the strips. Then, the actuation stress started to increase at the *T*_g_ transition between the glassy and the rubbery strip state as the raising temperature gradually released frozen arrangements of the epoxy crosslinked structure and concurrently longitudinally stretched and transversely compressed the nanotube structure of the MWCNT nanopaper. The recovered stress in the rubbery composite strips between the minimum and the maximum stress values plotted in Figure 3 was 1.5 and 0.88 MPa after their initial deformation of 2.71 and 1.36%, respectively. The analogous stress in the rubbery epoxy strips was 0.57 and 0.22 MPa after the initial deformation of 2.42 and 1.21%, respectively. Since the *T*_g_ temperature of 35.1 °C was exceeded in about 3 s after the start of heating, the stress in the epoxy strips regained rapidly. On the other hand, a stress recovery in the composite strips proceeded slightly slower, since the *T*_g_ temperature of 53.9 °C was not exceeded until 8 s after the start of heating. Obviously, the sooner the frozen actuation stress was released, the shorter was the stress recovery time.

The final actuation stress in the composite strips was two-times higher than the actuation stress in the epoxy strips at the corresponding strain pre-deformation, which corresponded to the ratio of their Young’s moduli as shown in Figure 4, since the stress and the strain of the composite and pure epoxy samples after reheating had the same values (Figure 3) as during the tensile tests (Figure 2 and Figure 4). The embedded nanopaper thus manifested as the reversibly elastically deformable element with a strong interlocking into an epoxy matrix responsible for the increase of the actuation force in the MWCNT nanopaper/epoxy composite.

After an extension of the composite over the nanopaper ultimate strain 0.8% (Figure 3), the embedded nanotube network could be partially disrupted, yet it still formed a conductive structure capable of both Joule heating of the composite during the tests of the reversible elastic deformation. Recently, we have observed partial cracks in the glassy MWCNT nanopaper/epoxy composite sensor integrated into the glass fiber/epoxy composite after its tensile straining of 0.94%, which was lower than the 2.9% fracture strain of the composite (Figure 5) [7]. The cracks ran perpendicularly to the direction of the extension. The resistance of such a cracked sensor was higher than of the intact one. This indicated that even though such damage had interrupted the nanotube network locally by a misconnection of the nanotubes, it did not fully disrupt the electrical conductivity of the embedded nanotube network, since an entanglement of carbon nanotubes and electrical conductance over the whole nanopaper remained.

The structural integrity of the nanopaper was tested only in compression. In the course of the compression, the nanopaper extended in the direction transverse to the direction of the compression force. The deformation of a material transverse to the direction of loading is called the Poisson effect. Its size is characterized by the Poisson’s ratio, which is the ratio of the positive transverse strain to the negative axial strain ν= −ε_transverse/_ε_axial_. The measured Poisson’s ratio ranged from 0.5 at a small compression to 0.29 at the final compression by the stress of 4 MPa, which induced a compressive strain of around 50%.

A SEM analysis of the nanopaper surface after the compressive deformation indicated that the integrity of the nanotube network was disrupted by randomly located cracks. These local cracks ran transverse to the extension direction and they were partially bridged by the remaining interconnected nanotubes (Figure 6). The view that cracks occur only locally is proved by a measurement of nanopaper electrical conductivity during compression/release cycles up to a maximum compressive stress of 8.7 MPa and a compressive strain of 55% [3].

In the second shape recovery test, the return of the bent MWCNT nanopaper/epoxy composite strip and the epoxy strip to their original straight shapes was assessed. After heating above *T*_g_, the straight strips were bent at a right angle, and once the samples cooled to the room temperature, this shape was fixed.

When the bent strip was reheated by Joule heating with an electric current of different actuating voltages, the frozen actuation stress in the embedded MWCNT nanopaper and the epoxy matrix was released, which drove the bent strip to restore to some degree its original straight shape. Depending on the final transformation temperature, the rearrangement of the deformed networks of the embedded nanotubes and the crosslinking of epoxy molecules was fully or partially restored to their arrangement before bending (Figure 7). Such a shape change was enumerated as a straightening ratio *S*_r_ (%) = (α_i_ − α_f_)/ α_i_ × 100, where α_i_ denotes the initial bending angle (90°) and α_f_ the final bending angle. At the final transformation temperature of 76.2 °C the straightening of the bent strip to the straightened one (*S*_r_ = 100%) was achieved in about 43 s. At 47.1 °C, i.e., at the temperature below *T*_g_ (53.6 °C), the final unbent angle 64° and the maximum *S*_r_ only 29% was achieved in about 600 s. The corresponding straightening speed, which was defined as *S*_r_^max^/*t*, where *S*_r_^max^ denoted the maximum straightening ratio, was *S*_r_^max^/*t* = 100/43 = 2.3 at the temperature of 76.2 °C and *S*_r_^max^/*t* = 29/621 = 0.05 at the temperature of 47.1 °C.

The released nanotube structure and the crosslinked epoxy molecular network at the temperature above *T*_g_ exerted a torque, which unbent the composite strip. Specifically, the strength of the unbending torque in the MWCNT nanopaper/epoxy composite straightened the strips from the right-angled shape bent down (Figure 8). The torque was varied by an addition of weights at the moving end of up to 4.5 g. Unlike the heating of the composite without weights (Figure 7), where the heating and the unbending started simultaneously, the MWCNT nanopaper/epoxy composite and the epoxy strips loaded with the weights were first heated to temperatures about 12 °C above their *T*_g_ (i.e., to 65 °C and 47 °C, respectively), and then the strips were released to unbend. The strip straightening ratio started with the full (i.e., 100%) straightening at the zero load and ended with the incomplete straightening of 49% at the maximum load of 4.5 g (Figure 8). The compressive deformation at the inner surface of the bent strip (the bend radius 5 mm) and the tensile deformation at the outer surface of the strip (thickness about 400 µm) were, according to the theory of simple beam bending, about 4%.

The straightening of the heated bent composite strips was compared to the straightening of the epoxy strips under comparable conditions, see Figure 9. The ability of the epoxy strip to regain the straight shape was less than that of the MWCNT nanopaper/epoxy composite strip. When the composite and epoxy strips were loaded with approximately the same weights of 4.5 and 4.1 g, respectively, the composite strip was able to achieve the straightening ratio of 49%, yet the epoxy strip merely achieved the straightening ratio of 6%. Alike the stress recovery in the pre-deformed straight strips (Figure 3), the frozen stress in the bent strips was released more rapidly in the epoxy strips than in the composite strips. In the first tests, the underlying mechanism was chiefly a faster surpassing of *T*_g_ in the epoxy strips (*T*_g_ = 35.1 °C) than in the composite strips (*T*_g_ = 53.9 °C), while in the second tests, the underlying mechanism was mainly affected by the entanglement of the epoxy segments and the carbon nanotubes of the porous nanopaper as discussed in the following Section. The differences found in the times and straightening ratio values of the strips with comparable loads suggested higher actuation torques for composite strips than for epoxy strips. On the other hand, the straightening of the composite strips proceeded more slowly than that of the epoxy strips.

The embedded nanopaper substantially increased the glass transition temperature *T*_g_ of the composite in comparison to the pure epoxy (53.9 °C and 35.1 °C, respectively). The DSC analysis after the physical aging of the epoxy and the MWCNT nanopaper/epoxy composite specimens suggested only *one enthalpy relaxation peak*, whose area represented the amount of energy involved in the specimen transition from the glassy state to the rubbery state. The physical aging was initiated by a change in the temperature of the pure epoxy and the composite from equilibrium at approximately *T*_g_ + 20 °C for 20 min to the annealing temperature *T*_g_ − 5 °C, which was reached by a cooling at the rate 10 °C/min. After the required period of aging *t*_a_ the samples were reheated at the rate of 10 °C/min to *T*_g_ + 20 °C. The obtained DSC curves and the peak temperature plot shown in Figure 10 and Figure 11 indicated an increase in the magnitude of the endothermic peak to higher temperature *T*_p_ as the aging time increased [21]. A shift in the peak temperatures of the composite may be affected by the entanglement of the epoxy segments and the carbon nanotubes of the porous nanopaper formed during the epoxy resin crosslinking. This issue is discussed in detail in the following Section. On the other hand, the enthalpy loss during the epoxy and composite physical ageing in the glassy state, as calculated by integrating the area under the DSC curve corresponding to the aging time, was not altered by the presence of the embedded nanopaper, see Figure 11.

## 4. Discussion

We embedded an electro-conductive MWCNT nanopaper in an epoxy matrix, and assessed its effect on the release of the frozen actuation force and the actuation torque after heating of the composite above its *T*_g_. At corresponding deformations, the recovered actuation stress in the composite strips was almost two-times higher than the recovered actuation stress in the pure epoxy strips, which corresponded to the ratio of their Young’s moduli as shown in Figure 2 and Figure 4. A more elastic composite, which requires a greater stress to deform, indicates a strong interlocking of nanopaper into an epoxy matrix. In the course of setting of a temporary shape of a rubbery epoxy strip, the segments between chemical crosslinks adapt to the external stretching load and elongate [19]. It results in a longitudinal orientation of most of the segments and a dislocation of crosslinked points [22]. Upon cooling and maintaining the deformed shape, secondary crosslinks are formed among the orientated segments. These secondary crosslinks are the main principle of the fixation of the molded shape [22]. After a reheating above *T*_g_, the shape recovery is initiated by detaching of the secondary crosslinks and releasing of the stored strain energy. After the elongation and the subsequent locking of the shape of the MWCNT nanopaper/epoxy composite strip by a cooling below *T*_g_ the embedded nanopaper was held in the stretched state by the frozen epoxy. Apparently, the individual nanotubes of the extended nanopaper network, which were intertwined with each other and with epoxy segments, were straightened longitudinally and their mechanical bonds shifted. When the nanopaper elongated longitudinally, it simultaneously compressed in the transverse direction. Successive cycles of loading and unloading causing a compressive deformation up to about 55% of a self-standing MWCNT nanopaper proved that the nanotube rearrangement is steady and reversible [3]. Similarly, reversible properties of the MWCNT nanopaper in tensile tests up to the tensile strain of 0.15% are presented in [23]. When the MWCNT nanopaper was subjected to compressive deformations, its integrity was compromised by local cracks as shown in Figure 6. When the nanopaper was embedded in an epoxy matrix, the embedding provided a support of the deformed nanopaper. Even at high nanopaper deformations, the MWCNT nanopaper/epoxy composite did not disintegrate at temperatures both below and above *T*_g_ and the nanotube network remained conductive, since it was still possible to use Joule heating to heat the composite as shown in Figure 7. When compared to the pure epoxy, the embedded nanopaper strengthened the frozen reversible actuation force in the composite as shown by a comparison of the straightening ratios in Figure 8 and Figure 9. Once the composite heated over the *T*_g_ to the transformation temperature of the shape memory, the epoxy matrix softened and the shape recovery stress in the embedded MWCNT nanopaper was released by detaching of the secondary crosslinks and by releasing of the stored epoxy strain energy of stretched segments within chemical crosslinks. At the same time, when the frozen actuation force in the epoxy was released, the deformed nanopaper nanotubes began to relax to their initial positions before deformation and thus the actuation force of the composite increased when compared to the pure epoxy.

The dependence of the release of the reversible force in the composite on the release of frozen polymer epoxy segments was confirmed by the measurement of different degrees of release of the frozen force at different temperatures (Figure 7). An interconnected network of polymer segments is constructed during the curing process of the epoxy and intensive molecular interactions between chemical groups on the carbon nanotube surface, epoxy and the curing agent may participate in the curing reaction, leading to the higher crosslinking density. When the liquid compounds infiltrated into the porous structure of a nanopaper with an average pore size of about 20 nm, the polymer segments intertwined with the carbon nanotubes of the nanopaper, and thus were constrained in the nanoscale interspaces, which restrained their mobility. This mechanical interconnection not only increased the strength of the rubbery MWCNT nanopaper/epoxy composite as discussed above, but it may also be the cause of a shift in the peak temperatures of the composite (Figure 10). While the polymer cross-linked polymer network in the pure epoxy is not restricted in its release during the glass/rubber transition, the constrained polymer network inside the pores of the nanopaper needs a higher energy that has to be added in the form of heat to release a frozen Brownian motion of the network segments [24]. Consequently, the strengthening mechanism of the electrothermal actuation was thus also affected, and the straightening of composite strips proceeded more slowly than that of the epoxy strips.

## 5. Conclusions

We proposed a measurement that allowed us to assess and analyze the strengthening mechanism of the electrothermal actuation by Joule heating in a composite consisting of a multiwalled carbon nanotube nanopaper embedded in an epoxy matrix. The measurement was performed by reheating of the straight glassy composite and pure epoxy strips, which held in the pre-stretched state by the frozen actuation stress, and by monitoring the release of the actuation stress in the strip samples during the glass/rubber transition. We found that the regained actuation stress in the composite strips was two-times higher than the similar stress in the pure epoxy strips at corresponding pre-stretched levels. This was because the more elastic structure of the stretched mutually intertwined epoxy segments and nanotubes of the nanopaper released when reheated a greater actuation stress than the less elastic stretched network of crosslinked segments of the pure epoxy.

## Figures and Tables

**Figure 1 nanomaterials-11-01529-f001:**
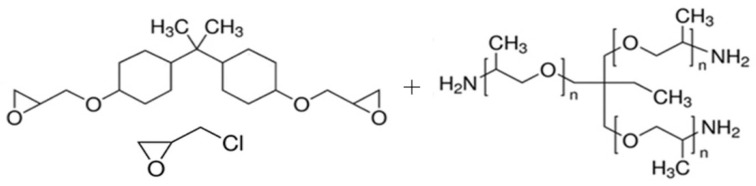
Structural formula of the component A—hydrogenated bisphenol A polymer with epichlorohydrin (left) and the component B—trimethylolpropane tris [poly(propylene glycol), amine terminated] ether.

**Figure 2 nanomaterials-11-01529-f002:**
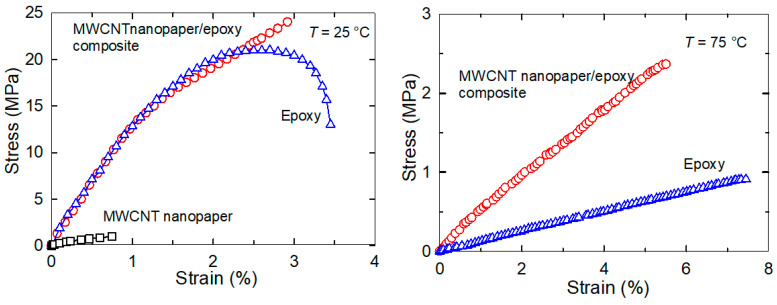
**Left** panel: tensile tests of the MWCNT nanopaper (squares), the glassy MWCNT nanopaper/epoxy composite (circles) and the glassy cured epoxy (triangles) at the elongation rate of 1 mm/min at the temperature 25 °C [18]. **Right** panel: tensile tests of the rubbery MWCNT nanopaper/epoxy composite (circles) and the rubbery epoxy (triangles) at the elongation rate of 1 mm/min at the temperature of 75 °C.

**Figure 3 nanomaterials-11-01529-f003:**
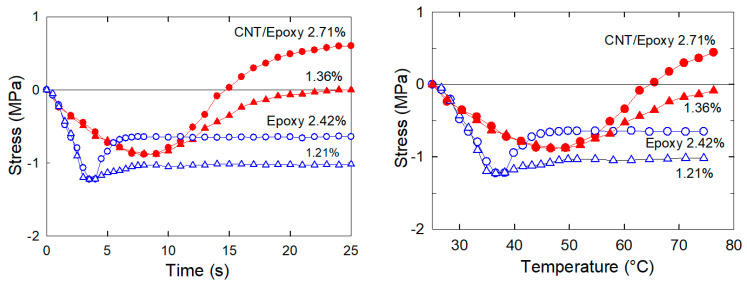
Time−dependent (**left**) and temperature−dependent (**right**) variations in stress during a heating of the MWCNT nanopaper/epoxy and epoxy strips from the temperature of 25 °C to 75 °C at the fixed strain denoted in the graphs.

**Figure 4 nanomaterials-11-01529-f004:**
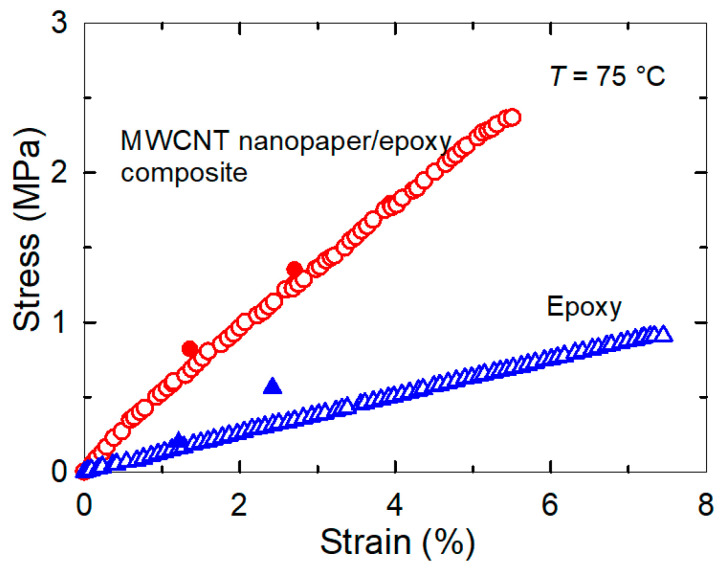
Comparison of the recovered stress after the reheating of the MWCNT nanopaper/epoxy composite (red filled circles) and the epoxy (blue filled triangles) strips to the temperature of 75 °C together with the results of the tensile test performed at the same conditions from Figure 3.

**Figure 5 nanomaterials-11-01529-f005:**
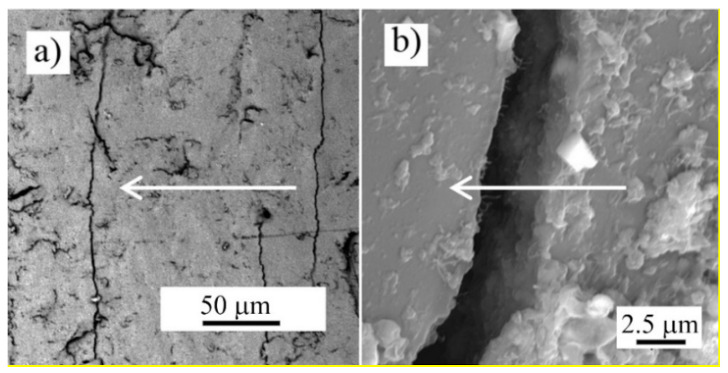
SEM micrograph of the surface of the MWCNT nanopaper/epoxy composite. (**a**) Cracks in the direction perpendicular to the direction of the strain (indicated by the arrow) appeared after the composite extension of 0.94% for 1 min at room temperature. (**b**) A detailed view of the crack.

**Figure 6 nanomaterials-11-01529-f006:**
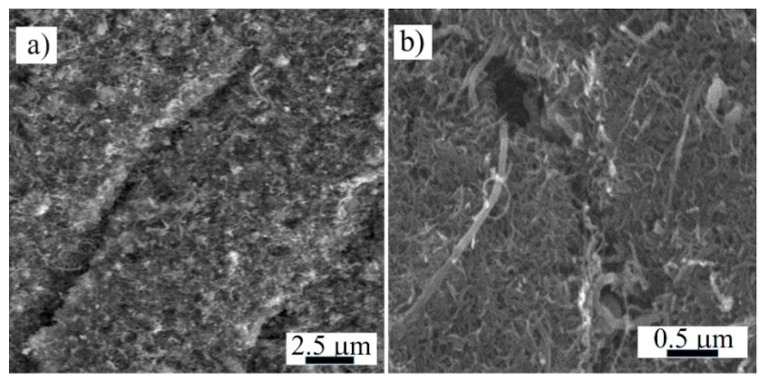
SEM micrographs of the surface of the entangled MWCNT network structure with cracks transverse to the direction of the extension of the nanopaper after its compression by 4 MPa. (**a**) A detailed view of the crack ending. (**b**) A view of nanotubes bridging a crack.

**Figure 7 nanomaterials-11-01529-f007:**
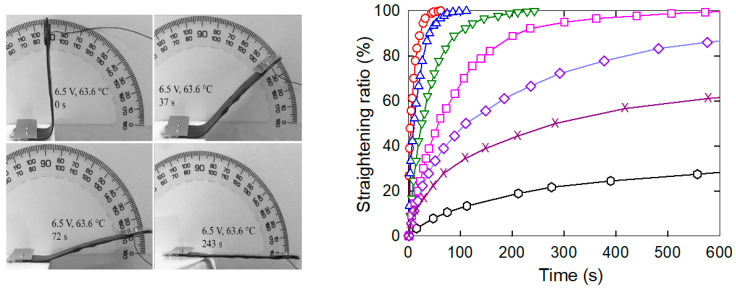
**Left**: The progress of the straightening of a bent strip over time indicated by green open triangles down in the adjacent plot. **Right**: The dependence of the strip straightening ratio on time in the course of Joule heating by 7.5 V from the temperature of 25 °C to 76.2 °C (red open circles), to 69.3 °C by 7 V (blue open triangles up), to 63.6 °C by 6.5 V (green open triangles down), to 58.5 °C by 6 V (pink open squares), to 55 °C by 5.5 V (dark blue open diamonds), to 50 °C by 5 V (dark pink X) and to 47.1 °C by 4.5 V (black open hexagons).

**Figure 8 nanomaterials-11-01529-f008:**
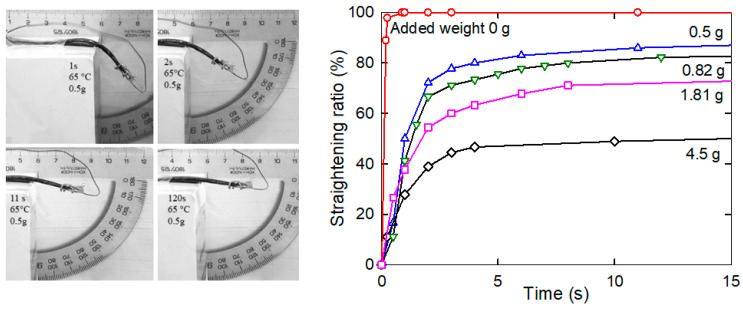
**Left**: the depiction of the incomplete straightening of the heated bent composite strip loaded with a weight of 0.5 g at indicated times and at a temperature of 65 °C (the entire time course is depicted in the graph with blue open triangles in the adjacent plot). **Right**: the dependence of the composite strip straightening ratio on time at 65 °C when unloaded (denoted as 0 g, red open circles) or loaded with weights 0.5 g (blue open triangles up), 0.82 g (green open triangles down), 1.81 g (pink open squares) and 4.5 g (dark blue open diamonds).

**Figure 9 nanomaterials-11-01529-f009:**
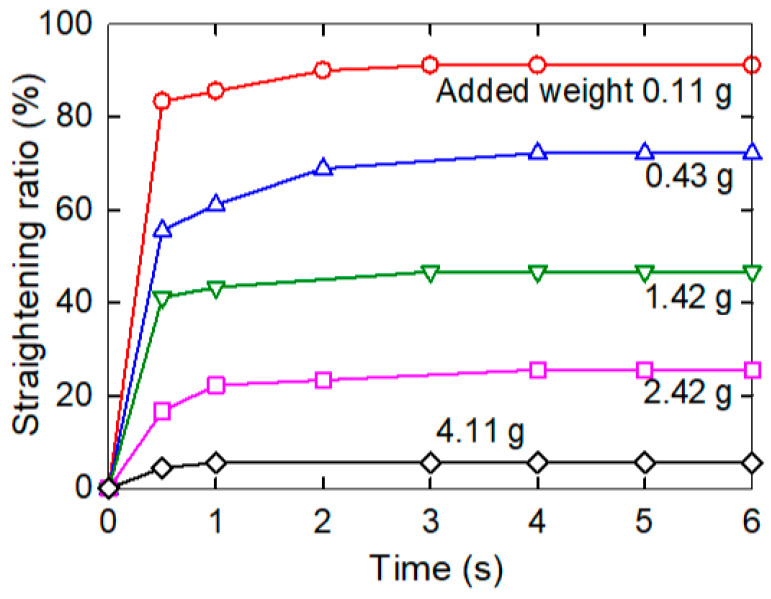
The time course of the straightening of the epoxy strips loaded with indicated weights at 47 °C.

**Figure 10 nanomaterials-11-01529-f010:**
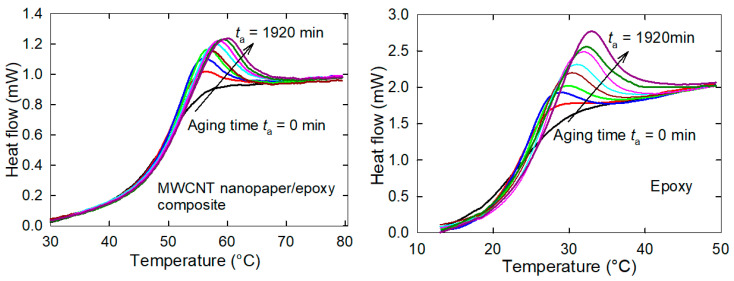
DSC curves for the MWCNT nanopaper/epoxy composite and the pure epoxy obtained when samples were heated after the aging time *t*_a_ = 0, 15, 30, 60, 120, 240, 480, 960, 1920 min. A stepwise increase in the aging time is indicated by the arrow.

**Figure 11 nanomaterials-11-01529-f011:**
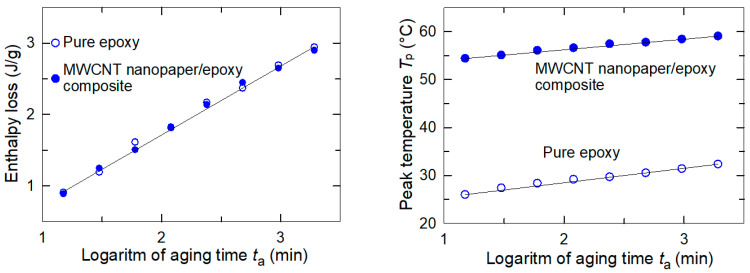
The enthalpy loss and the peak temperature *T*_p_ dependence on aging time.

## Data Availability

The data is available upon reasonable request from the corresponding author.

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
