# Peer review of "Strengthening Mechanism of Electrothermal Actuation in the Epoxy Composite with an Embedded Carbon Nanotube Nanopaper"

_nanomaterials, 2021, doi:10.3390/nano11061529_

Round 1

Reviewer 1 Report

In this paper, the authors study the effect of an electro-conductive multiwalled carbon nanotube nanopaper embedded in an epoxy matrix and the relative behaviour during the heating of the composite. To measure and analyze the mechanism of electrothermal actuation controlled by the heating with an electric current of different voltages, a specific procedure was proposed.

In general, the manuscript could be interesting if improved in the basic aspects. In particular, the authors use a commercial product as epoxy matrix, without providing information about the chemical structure of the epoxide and the crosslinking agent. In my opinion, the knowledge of the chemical structures and the exact amount of the epoxy resin and of the curing agent is very important to correlate the experimental data with the composite composition.  Details about the chemical structure could help to understand if a chemical interaction between nanopaper and epoxy matrix (resin and curing agent) occurs. Also, the knowledge of the exact epoxy resin/curing agent/nanopaper amount allows determining the thermal phenomena observed through DSC experiments. Instead, the author wrote a general sentence “…the nanopaper was laid on a polytetrafluorethylene (PTFE) foil and filled up with the epoxy resin.”. I think that the experimental conditions must be explained more deeply. So, in my opinion, the manuscript cannot be evaluated in this form and must be improved taking account of the above issues, and then resubmitted.

Author Response

 Reply - Reviewer #1

 Thank you very much for your comments to our article entitled ”Strengthening mechanism of electrothermal actuation in the epoxy composite with an embedded carbon nanotube nanopaper” by P. Slobodian et al. submitted to Nanomaterials. We have duly considered all of them and have found the remarks and suggestions very constructive and acceptable. Consequently, the revised version of the paper takes into accounts all of them. The corresponding responses to your comments are below.

Comment1

In particular, the authors use a commercial product as epoxy matrix, without providing information about the chemical structure of the epoxide and the crosslinking agent.

Reply:

Thank you for the comment. We have enlarged the part providing information about the chemical structure of the epoxide and the crosslinking agent by almost one page. Please, see the fourth page of the article.

Comment 2

Also, the knowledge of the exact epoxy resin/curing agent/nanopaper amount allows determining the thermal phenomena observed through DSC experiments. Instead, the author wrote a general sentence “…the nanopaper was laid on a polytetrafluorethylene (PTFE) foil and filled up with the epoxy resin.”. I think that the experimental conditions must be explained more deeply.

Reply:

Thank you for the comment. We inserted into the mentioned text the detailed data on the epoxy resin/curing agent ratio in the mixture (Section 2.1 page 4) and its volume (Section 2.1 page 5), which filled the pores of the nanopaper. Inserted specifications are marked in yellow.

…..Next, the porous nanopaper was laid on a polytetrafluorethylene (PTFE) foil and its pores were filled up with ca. 0.2 ml of the above described epoxy resin and curing agent at the A:B ratio 100:75. After filling, the nanopaper permeated by the epoxy resin and curing agent mixture was covered with another PTFE foil and loaded with a pressure of 300 kPa to avoid wrinkling in the course of curing by Joule heating at temperature around 60 °C for 15 min. …..

As far as the knowledge of the exact epoxy resin/curing agent/nanopaper interactions from molecular point of view and determination the thermal phenomena observed through DSC experiments is concerned, we consider the molecular interaction side of the problem to be very important and crucial. Consequently, details are newly given in Section 2.1, page 4, paragraph 2) and Section 4, page 17, where we added the following text marked in yellow:

……An interconnected network of polymer segments is constructed during the curing process of the epoxy and intensive molecular interactions between chemical groups on the carbon nanotube surface, epoxy and the curing agent may participate in the curing reaction, leading to the higher crosslinking density. When the liquid compounds infiltrated into the porous structure of a nanopaper with an average pore size of about 20 nm, the polymer segments intertwined with the carbon nanotubes of the nanopaper and thus were constrained in the nanoscale interspaces, which restrained their mobility. ……

Thermal phenomena observed through DSC experiments are interpreted in the article based on the interaction of polymer segments and carbon nanotubes, as  (Section 3, page 15):

…The obtained DSC curves and the peak temperature plot shown in Figs. 9 and 10 indicated an increase in the magnitude of the endothermic peak to higher temperature Tp as the aging time increased [21]. A shift in the peak temperatures of the composite may be affected by the entanglement of the epoxy segments and the carbon nanotubes of the porous nanopaper formed during the epoxy resin crosslinking. ….

More details on the epoxy curing and thermal phenomena observed through DSC experiments could be revealed by the molecular dynamics simulation method such as molecular intraction between the individual reactans and their influence on the polymerization and interlocking of polymer segments and nanotubes. Since the aim of the article was to determine and describe the strengthening mechanism of electrothermal actuation in the epoxy composite with an embedded carbon nanotube nanopaper, we think that molecular dynamic simulation analysis is beyond the scope of this article.

Needless to say that thanks to your well-aimed comments, we were able to correct our manuscript. Thank you very much.

Yours sincerely,             P. Slobodian

Reviewer 2 Report

in this research, the author analyzed the effect of the release of the frozen actuation force and the actuation torque about MWCNT nanopaper/epoxy composite and other samples. SEM, thermal camera Flir E5-X, Fast-scanning data logger and other characterization methods are used to evaluate the performance of composite materials.

I considered it can be published in our journal after some revision:

  1. How to get the rubbery MWCNT nano-paper/epoxy resin composite and epoxy Young's modulus to be 46 MPa and 24 MPa? The manuscript could not find relevant data.
  2. “On the other hand, the MWCNT nanopaper/epoxy composite cured by the Joule heating at 4.4 V and the terminal temperature of 61 °C strengthened until the fracture stress of 24 MPa at the strain 2.9%”, is there a characterization to illustrate the conclusion?
  3. Why was the ability of the epoxy strip to regain the straight shape lesser than that of the MWCNT nanopaper/epoxy composite strip?
  4. As shown in Fig. 7, 8, and 9, there are two patterns and the order of labeling should be marked.
  5. The text of the article has different font sizes and should be unified. The scale of the SEM images are not standardized, and the font of the font size is not uniform.
  6. There are some grammar errors in the manuscript. The author should add the referees in the introduction or characterized part such as “Composites Part B, 2021, 205: 108529., Journal of Colloid and Interface Science 2021, 586: 479-490.”
  7. The horizontal and vertical coordinates of all legends in the manuscript should be centered, and the unit is written in parentheses.
  8. The language in the article needs to be carefully checked, especially the introduction, which should be modified.

Author Response

Reply - Reviewer #2

 Thank you very much for your comments to our article manuscript ID: Nanomaterials 1130069 entitled ”Strengthening mechanism of electrothermal actuation in the epoxy composite with an embedded carbon nanotube nanopaper” by P. Slobodian et al. submitted to Nanomaterials. We have duly considered all of them and have found the remarks and suggestion very constructive and acceptable. Consequently, the revised version of the paper takes into accounts all of them. The corresponding responses to your comments are below.

Comment1

     How to get the rubbery MWCNT nano-paper/epoxy resin composite and epoxy Young's modulus to be 46 MPa and 24 MPa? The manuscript could not find relevant data.

Reply:

Thank you for the comment. We have improved the text of the revised paper by inserting a Young modulus definition, marked in yellow in the sentence:

…. The strain dependency of stresses for the rubbery MWCNT nanopaper/epoxy composite and the epoxy was linear and the Young's moduli, which are defined as E = s/e (where s denotes tensile stress and e axial strain) were 46 MPa and 24 MPa, respectively. …. (Section 3, page 7, 1st paragraph).

Comment 2

“On the other hand, the MWCNT nanopaper/epoxy composite cured by the Joule heating at 4.4 V and the terminal temperature of 61 °C strengthened until the fracture stress of 24 MPa at the strain 2.9%”, is there a characterization to illustrate the conclusion?

Reply:

Thank you for the comment. To better illustrate the determination of the stated values of fracture stress and strain, we have extended the sentence by reference to Fig. 1 (marked in yellow), where these values are determined as the final values of the usual curves for tensile tests of the corresponding samples. In the case of a MWCNT nanopaper/epoxy composite, the curve is indicated by open red circles and the fracture stress and strain are the maximum values of stress and strain. The new version of the sentence is:

…… On the other hand, the MWCNT nanopaper/epoxy composite cured by the Joule heating at 4.4 V and the terminal temperature of 61 °C strengthened until the fracture stress of 24 MPa at the strain 2.9% as seen in Fig. 1. ….

Comment 3

Why was the ability of the epoxy strip to regain the straight shape lesser than that of the MWCNT nanopaper/epoxy composite strip?

Reply:

This issue is discussed in great detail (nearly 2 pages) in Section 4. Among other things, it states there that:

…… When compared to the pure epoxy, the embedded nanopaper strengthened the frozen reversible actuation force in the composite. Once the composite heated over the Tg to the transformation temperature of the shape memory, the epoxy matrix softened and the shape recovery stress in the embedded MWCNT nanopaper was released by detaching of the secondary crosslinks and by releasing of the stored epoxy strain energy of stretched segments within chemical crosslinks. At the same time, when the frozen actuation force in the epoxy was released, the deformed nanopaper nanotubes began to relax to their initial positions before deformation and thus the actuation force of the composite increased when compared to the pure epoxy. Etc. …

Comment 4

As shown in Fig. 7, 8, and 9, there are two patterns and the order of labeling should be marked.

Reply:

At first, we apologize if did not understand your comment well. According to how we understood it, we have inserted into Figs. 7 and 8 a description indicating the inserted numbers as the values of the added weights on at the moving end of the unbending composite strips to increase the strength of the unbending torque. As far as the Fig. 9 concerns, we have inserted the values of the first and the final aging times into the graph.

Comment 5

The text of the article has different font sizes and should be unified. The scale of the SEM images are not standardized, and the font of the font size is not uniform.

Reply:

Thank you for the comment. Text fonts have been unified and the scale of SEM images has been standardized and font size unified.

Comment 6

There are some grammar errors in the manuscript. The author should add the referees in the introduction or characterized part such as “DOI: 10.1016/j.cej.2021.128875., Composites Part B, 2021, 205: 108529., Composites Part B, 2021, 204: 108491., Chemical Engineering Journal, 2020, 402: 125951., Journal of Colloid and Interface Science 2021, 586: 479-490.”

Reply:

Grammatical errors were corrected. However, we are sorry to say that we do not understand the meaning of the next part of your comment. This does not seem to be the case for our article.

Comment 7

The horizontal and vertical coordinates of all legends in the manuscript should be centered, and the unit is written in parentheses.

Reply:

The horizontal and vertical coordinates of all legends in the manuscript were centered and the units are enclosed in parentheses as required.

Comment 8

The language in the article needs to be carefully checked, especially the introduction, which should be modified.

Reply:

Thank you very for the comment. English was thoroughly multiple times checked in the revised paper version and we hope that from this point of view the text will be acceptable. The Section Introduction as well as the Abstract have been partially rewritten.

Needless to say, thanks to your well-founded comments we were able to modify our manuscript considerably. Thank you very much indeed.

Yours sincerely,             P. Slobodian

Round 2

Reviewer 1 Report

In my opinion, considering the last amended version, the manuscript has been improved. Nevertheless, the manuscript has several typos and mistakes.

Some examples:

- pictures in figure 2 are inverted and not well discussed in the text

- Figure 4 (left) shows meaningless numbers overlapped to the picture

- Figure 5 (left) shows meaningless numbers overlapped to the picture

- Figure 7 (left) shows meaningless numbers overlapped to the picture

- Maint text (and captions) seems to have different fonts and sizes (as an example, see pages 6, 7, 9, etc.)

Moreover:

- the chemical structure of Epox G 200 components must be reported in a figure

- The text reported in the experimental section  “The poly(propylene glycol) chains themselves are very movable, capable to rotate around a single chemical bond. The chains contain oxygen, which has a relatively low rotational barrier and therefore the conformation of the chains can change easily under, e.g. an applied mechanical stress. The reaction of the two components A and B leads to a formation of a three-dimensional polymer network of a final epoxy matrix. At the A:B ratio 100:100, an excess of the component B causes that some poly(propylene glycol) branches remain unbounded because of a lack of oxirane rings with a high molecular mobility potential, and the epoxy is elastic. At the ratio 100:50, the proportion of functional groups for crosslinking between components A and B is more equimolecular. The epoxy contains less free unbounded poly(propylene glycol) branches, which results in a resistance against conformation changes, stiffness and a higher Tg. According to the supplier, the ratio of the epoxy components (A/B-hard/elastic) 100:50 yields a hard epoxy of hardness 79 Shore D, while the component ratio of 100:100 yields a flexible one (44 Shore A).” should be improved and moved in the right place within the main text

- All the figures should be better discussed in the text

In conclusion, I suggest to the authors a deep check of the manuscript, because in this form it is not suitable for publication.

Author Response

Thank you very much for your comments to our article entitled ”Strengthening mechanism of electrothermal actuation in the epoxy composite with an embedded carbon nanotube nanopaper” by P. Slobodian et al. submitted to Nanomaterials. They help us a lot to edit and improve our article, for which we thank you very muchonce more.All remarks and suggestions arevery constructive. The corresponding responses to your comments are below.

Comment 1

Pictures in figure 2 are inverted and not well discussed in the text.

Reply:

Thank you for the comment. The order of the graphs in Fig. 2 has been adjusted.The discussion ofthe results shown in Fig. 2 wassupplemented by text and references to specific figuresmarked in yellow as follows.

…..The recovered stress in the rubbery composite strips between the minimum and the maximum stress values plotted in Fig. 2 was 1.5 and 0.88 MPa after their initial deformation of 2.71 and 1.36%, respectively. The analogous stress in the rubbery epoxy strips was 0.57 and 0.22 MPa after the initial deformation of 2.42 and 1.21%, respectively. Since the Tgtemperature of 35.1 °C was exceeded in about 3 seconds after the start of heating, the stress in the epoxy strips regained rapidly. On the other hand, a stress recovery in the composite strips proceeded slightly slower, since the Tgtemperature of 53.9 °C was not exceeded until 8 s after the start of heating. Obviously, the sooner the frozen actuation stress was released, the shorter was the stress recovery time.

The final actuation stress in the composite strips was two-times higherthan the actuation stress in the epoxy strips at the corresponding strain pre-deformation, which corresponded to the ratio of their Young'smoduli as shown in Fig. 3, since the stress and the strain of the composite and pure epoxy samples after reheating had the same values (Fig. 2) as during the tensile tests (Figs. 1 and 3). The embedded nanopaper thus manifested as the reversibly elastically deformable element with a strong interlocking into and epoxy matrix responsible for the increase of the actuation force in the MWCNT nanopaper/epoxy composite. ....

In addition, the entire Section 4. (Discussion) is devoted on two pages to the discussion of results shown in Figs.1-3 and 6-9 and especially their mutual interconnection to show the role and importance of a nanopaper for the strengthening mechanism of electrothermal actuation in the epoxy composite with an embedded carbon nanotube nanopaper.

Comment 2

Figure 4 (left) shows meaningless numbers overlapped to the picture.Figure 5 (left) shows meaningless numbers overlapped to the picture.

Reply:

Inset scale bars into the SEM micrographs in the Figures 4 and 5 are graphical means of depicting the lengthmeasureand have a label showing the actual length of the barbefore being magnified. The scale bars in the Figures 4 and 5 arenecessaryto indicate the length rate of the cracks shown in the micrographs. For this reason, we leave the Figures 4 and 5 unchanged.

Comment 3

Figure 7 (left) shows meaningless numbers overlapped to the picture.

Reply:

Thank you for the comment. To emphasize the inlet numbersin the photographs, we have modified the figure caption accordingly.Modifications are marked in yellow.

Fig. 7 Left: the depiction of the incomplete straightening of the heated bent composite strip loaded with a weight of 0.5 g at indicated times and at a temperature of 65 °C (the entire time course is depicted in the graph with blue open triangles in the adjacent plot). ...

Comment 4

Maint text (and captions) seems to have different fonts and sizes (as an example, see pages 6, 7, 9, etc.

Reply:

Thank you very much for the comment. The fonts and their sizes were carefully checkedaccordingly. The only error found on page 9 for Greek letters has been corrected.

Comment 5

The chemical structure of Epox G 200 components must be reported in a figure.

Reply:

The chemical structure of Epox G 200 components is shown in the new Fig. 1.

Comment 6

The text reported in the experimental section “The poly(propylene glycol) chains themselves are very movable, capable to rotate around a single chemical bond. The chains contain oxygen, which has a relatively low rotational barrier and therefore the conformation of the chains can change easily under, e.g. an applied mechanical stress. The reaction of the two components A and B leads to a formation of a three-dimensional polymer network of a final epoxy matrix. At the A:B ratio 100:100, an excess of the component B causes that some poly(propylene glycol) branches remain unbounded because of a lack of oxirane rings with a high molecular mobility potential, and the epoxy is elastic. At the ratio 100:50, the proportion of functional groups for crosslinking between components A and B is more equimolecular.

The epoxy contains less free unbounded poly(propylene glycol) branches, which results in a resistance against conformation changes, stiffness and a higher Tg. According to the supplier, the ratio of the epoxy components (A/Bhard/elastic) 100:50 yields a hard epoxy of hardness 79 Shore D, while the component ratio of 100:100 yields a flexible one (44 Shore A).” should be improved and moved in the right place within the main text.

Reply:

The mentioned text has been moved to the beginning of Section 3. Results.

Comment 7

All the figures should be better discussed in the text.

Reply:

Thank you very much for mentioning issue, whichis quite important for us.We are fully aware of this perhaps unconventional approachof not discussing in details the experimental results continuously in the text. But individual measurements are systematically performed in order to be able to use them for the basic aim of the manuscript, namely the explanation of the strengtheningmechanism of electrothermal  actuation in  the  epoxy composite  with an embedded carbon nanotube nanopaper. We therefore believe that it is right to discuss particular measurements together in one part of the manuscript, namely in the two pages long Section 4 (Discussion), where the experimental results can be linked into a logical complex and not discussed separately, which would not make their interconnection and documentation capability so obvious.

Thank you very much once more for thewell-aimed commentsthat help us to improveour manuscript.

Yours sincerely,       P. Slob

Round 3

Reviewer 1 Report

The manuscript has been improved, but it must be checked carefully because there are some mistakes inside the text (i.e. different fonts or fonts size, etc..)

Author Response

Reply - Reviewer #1

     Thank you very much for your comment to our article entitled ”Strengthening mechanism of electrothermal actuation in the epoxy composite with an embedded carbon nanotube nanopaper” by P. Slobodian et al. submitted to Nanomaterials. The corresponding response to your comment is below.

Comment1

The manuscript has been improved, but it must be checked carefully because there are some mistakes inside the text (i.e. different fonts or fonts size, etc..)

Reply:

Thank you for the comment. We re-read the manuscript very carefully and made the appropriate changes to the fonts and their sizes. The corrections are marked up using the “Track Changes” function so that they are easily recognizable. 

This manuscript is a resubmission of an earlier submission. The following is a list of the peer review reports and author responses from that submission.